# Prediction of Mortality in Geriatric Traumatic Brain Injury Patients Using Machine Learning Algorithms

**DOI:** 10.3390/brainsci13010094

**Published:** 2023-01-03

**Authors:** Ruoran Wang, Xihang Zeng, Yujuan Long, Jing Zhang, Hong Bo, Min He, Jianguo Xu

**Affiliations:** 1Department of Neurosurgery, West China Hospital, Sichuan University, 610041 Chengdu, China; 2Department of Critical Care Medicine, Chengdu Seventh People’s Hospital, 610021 Chengdu, China; 3Department of Critical Care Medicine, West China Hospital, Sichuan University, 610041 Chengdu, China

**Keywords:** traumatic brain injury, geriatric, machine learning, prognosis, prediction

## Abstract

Background: The number of geriatric traumatic brain injury (TBI) patients is increasing every year due to the population’s aging in most of the developed countries. Unfortunately, there is no widely recognized tool for specifically evaluating the prognosis of geriatric TBI patients. We designed this study to compare the prognostic value of different machine learning algorithm-based predictive models for geriatric TBI. Methods: TBI patients aged ≥65 from the Medical Information Mart for Intensive Care-III (MIMIC-III) database were eligible for this study. To develop and validate machine learning algorithm-based prognostic models, included patients were divided into a training set and a testing set, with a ratio of 7:3. The predictive value of different machine learning based models was evaluated by calculating the area under the receiver operating characteristic curve, sensitivity, specificity, accuracy and F score. Results: A total of 1123 geriatric TBI patients were included, with a mortality of 24.8%. Non-survivors had higher age (82.2 vs. 80.7, *p* = 0.010) and lower Glasgow Coma Scale (14 vs. 7, *p* < 0.001) than survivors. The rate of mechanical ventilation was significantly higher (67.6% vs. 25.9%, *p* < 0.001) in non-survivors while the rate of neurosurgical operation did not differ between survivors and non-survivors (24.3% vs. 23.0%, *p* = 0.735). Among different machine learning algorithms, Adaboost (AUC: 0.799) and Random Forest (AUC: 0.795) performed slightly better than the logistic regression (AUC: 0.792) on predicting mortality in geriatric TBI patients in the testing set. Conclusion: Adaboost, Random Forest and logistic regression all performed well in predicting mortality of geriatric TBI patients. Prognostication tools utilizing these algorithms are helpful for physicians to evaluate the risk of poor outcomes in geriatric TBI patients and adopt personalized therapeutic options for them.

## 1. Introduction

Population aging is a challenge in most of the developed countries. Estimated by the American Census Bureau, the elderly population (age ≥ 65 years) in the United States will increase to 80 million by 2050 [1]. The elderly population in Japan and South Korea has, respectively, reached to 27.7% and 13.8% in 2017 [2,3]. And the trend of population aging will remain or even be enhanced in the next decades. With the increase of the elderly population, the number of elderly traumatic brain injury (TBI) patients is also gradually increasing. It has been reported that emergency department visits and hospitalizations for TBI in elderly people of United States increased by 46% and 34%, respectively [4]. A report analyzed from the Japan Neurotrauma Data Bank Project 2015 indicated that 53.6% of registered TBI patients were elderly (age ≥ 65 years) and that most severe TBI patients were elderly [5]. Additionally, impaired performance of muscle strength, balance and agility caused by aging render older adults more likely to fall than young people [6]. Actually, more than half of TBI incidents among the elderly are attributable to ground-level falls [7].

Previous studies have shown that age is an independent risk factor of TBI prognosis [8,9]. And elderly TBI patients commonly suffer more complications and unfavorable outcomes than do non-elderly TBI patients [10,11]. Research from different countries has reported that the mortality rate of geriatric TBI ranged from 6.4% to 67.2% [3,12,13,14,15]. Although some elderly TBI patients do not suffer death in the short term, these TBI survivors survive with prominent physical and cognitive deficits [16]. Additionally, TBI survivors commonly develop psychiatric disorders and tend to be at higher risk of dementia and Alzheimer’s [17,18,19]. These disabilities and sequelas would continuously affect quality of life, and they bring a heavy economic burden for geriatric TBI patients [20,21]. Therefore, evaluating the prognosis of geriatric TBI patients early on could guide doctors in making individualized treatments and rehabilitation strategies for improving the prognosis, quality of life and reducing the medical expenditure.

Many previous studies have developed prognostic models for geriatric TBI utilizing conventional logistic regression [8,22,23,24]. Some risk factors for poor prognosis have been found, such as age, Charlson Comorbidity Index, Glasgow Coma Scale (GCS), Injury Severity Score (ISS), systolic blood pressure, intraventricular hemorrhage, and neurosurgical intervention [8,22,23,24]. However, there are no studies using machine learning algorithms to evaluate the prognosis of geriatric TBI. Compared with the conventional logistic regression, machine learning algorithms may perform better in analyzing nonlinear correlations and handling massive high-dimensional datasets. We designed this study to explore the prognostic value of different machine learning algorithm-based models for predicting mortality in geriatric TBI patients.

## 2. Materials and Methods

### 2.1. Patients

Patients included in this study were found in the Medical Information Mart for Intensive Care-III (MIMIC-III) database designed and produced by the computational physiology laboratory of Massachusetts Institute of Technology (MIT) (Cambridge, MA). This freely available database collects the information of patients admitted to Beth Israel Deaconess Medical Center (BIDMC) (Boston, MA) between 2001 and 2012 and obtains pre-approval from the institutional review boards of MIT and BIDMC. All patients included in the MIMIC-III were deidentified and anonymized in consideration of privacy protection. We included patients with head injury from the MIMIC-III based on ICD-9 codes (80000–80199; 80300–80499; 8500–85419). Then, patients were excluded according to the following criteria: (1) TBI patients with age < 65; (2) patients who lacked records of GCS on admission; (3) Abbreviated Injury Score (AIS) head < 3; or (4) patients who lacked records of vital signs and laboratory test (Figure 1). After screening, 1123 patients were finally included in the study.

### 2.2. Data Collection

Age, gender, and comorbidities, including diabetes mellitus and hypertension were collected. Records of vital signs on admission, including systolic blood pressure, diastolic blood pressure, heart rate, respiratory rate, body temperature, and pulse oxygen saturation (SpO_2_) were extracted. Clinical scores including GCS, AIS of face, head, chest, abdomen, surface, and limb, and ISS were included [25,26]. Anatomical intracranial injury locations including epidural hematoma, subdural hematoma, subarachnoid hemorrhage, and intracerebral hemorrhage were classified based on ICD-9 codes. The results of laboratory tests analyzed from the first blood sample after admission were extracted, including white blood cell, platelet, red blood cell, red cell distribution width, hemoglobin, glucose, blood urea nitrogen, serum creatinine, sodium, potassium, phosphorus, calcium, magnesium, chloride, anion gap, prothrombin time, and international normalized ratio. Medical interventions including mechanical ventilation and neurosurgical operation were included. The primary outcome of this study was 30-day mortality. All above mentioned variables were extracted from the MIMIC-III through Navicat Premium 12 using Structure Query Language.

### 2.3. Statistical Analysis

The normality of included variables was confirmed by the Kolmogorov–Smirnov test. Normal distributed and non-normal distributed variables were presented as mean ± standard deviation and median (interquartile range), respectively. Categorical variables were shown as counts (percentage). Differences between the two groups of normal distributed and non-normal distributed variables were verified by Student’s t-test and the Mann–Whitney U test, respectively. A chi-square test or Fisher exact test was conducted to analyze the difference between two groups of categorical variables. To develop and validate machine learning algorithms-based models, all TBI patients were randomly divided between a training set (70%) and a testing set (30%). Logistic regression and six machine learning algorithms, including decision tree, Random Forest, support vector machine (SVM), Naïve Bayes, Adaboost and XGboost, were utilized to train predictive models for a 30-day mortality in training dataset. Variables with *p* < 0.05 in univariate logistic regression analysis were included into multivariate logistic regression analysis in the training set. The receiver operating characteristic (ROC) curve was drawn and the area under the ROC curve (AUC) was calculated to compare predictive performance of different machine learning algorithms-based models. Additionally, sensitivity, specificity, accuracy and F1 score (F1 score is calculated as the harmonic average of the precision rate and recall rate) were also calculated to evaluate the performance of these models.

All analyses were performed using R software (version 3.6.1; R Foundation, R Core Team, Vienna, Austria). R packages used for machine learning included ‘rpart’, ‘rpart.plot’, ‘party’, ‘randomForest’, ‘e1071′, ‘adabag’, and ‘xgboost’.

## 3. Results

### 3.1. Baseline Characteristics of Included TBI Patients

1123 TBI patients from the MIMIC-III were ultimately included, with a 30-day mortality of 24.8% (Table 1). Compared with survivors, non-survivors had higher age (*p* = 0.010) but lower incidence of hypertension (*p* = 0.007). Non-survivors had lower systolic blood pressure (*p* = 0.014), lower body temperature (*p* < 0.001) and higher SpO_2_ (*p* < 0.001) than survivors. Pupillary nonreactivity was more frequently observed in non-survivors (*p* < 0.001). Non-survivors had lower GCS (*p* < 0.001), higher AIS head (*p* < 0.001), AIS chest (*p* = 0.029), ISS (*p* < 0.001) and higher incidence of epidural hematoma (*p* = 0.001), and subarachnoid hemorrhage (*p* = 0.048). Results of laboratory tests showed that white blood cell (*p* < 0.001), red cell distribution width (*p* < 0.001), glucose (*p* < 0.001), blood urea nitrogen (*p* < 0.001), serum creatinine (*p* < 0.001), anion gap (*p* = 0.002), prothrombin time (*p* < 0.001), and international normalized ratio (*p* < 0.001) were higher in non-survivors, while platelet (*p* = 0.030), hemoglobin (*p* = 0.010), and calcium (*p* = 0.002) were lower in non-survivors. Finally, the usage incidence of mechanical ventilation was significantly higher in non-survivors (*p* < 0.001). Non-survivors had shorter length of hospital stay than survivors (*p* < 0.001).

### 3.2. Performance of Machine Learning Algorithms for Predicting Mortality in Geriatric TBI Patients

The AUC, sensitivity, specificity, accuracy and F score of machine learning algorithms for predicting mortality in the training set and the testing set are presented in Table 2. In the training set, Random Forest, Adaboost and XGboost reached the highest AUC of 1.000. In testing set, however, Adaboost, Random Forest and logistic regression ranked first, second and third, with AUC of 0.799, 0.795 and 0.792, respectively. ROC curves of machine learning algorithms for predicting mortality in the training set and the testing set are shown as Figure 2a,b. The importance of the top-20 features for predicting mortality in training set is shown in Figure 3a,b. The three most important features in Adaboost were body temperature, systolic blood pressure and white blood cell, sequentially. The three most important features in the Random Forest were GCS, AIS head and white blood cell, sequentially. The details of each variable in the logistic regression-based model was presented as Table 3.

## 4. Discussion

The 30-day mortality of included geriatric TBI patients in this study was 24.8%, which was similar to previously reported incidence ranging from 6.4% to 67.2% [3,12,13,14,15]. The significant mortality difference in different studies may be attributable to differences of injury severity, therapeutic options, and age distribution. A total of eight factors were found to be independently associated with mortality by the logistic regression, including age, body temperature, pupillary nonreactivity, GCS, AIS head, white blood cell, calcium, and mechanical ventilation, all of which have been confirmed as risk factors for poor prognosis in TBI.

Many previous studies have verified that increasing age was actually the strongest predictor of poor outcome in TBI [27,28,29]. The increase in age may indicate worse nutritional status, extracranial organ function, cerebrovascular autoregulation and higher likelihood of infectious complication, or secondary brain injury. The pupillary nonreactivity implying impaired function of medulla oblongata and midbrain, has been confirmed as an important and convenient index to evaluate the prognosis of TBI [30,31,32]. Although the GCS has been utilized to evaluate the condition of brain injury patients for decades, it shows unstable performance under several situations, including drinking, seizure, and being sedated. Especially, geriatric patients commonly suffer complications with cerebrovascular disease, dementia, and impaired hearing, which could limit the reliability of GCS evaluation [33]. The median GCS of our included geriatric TBI patients was 14, with a lower and upper quartile of 7 and 15, which indicates that most of included geriatric patients suffered mild to moderate TBI. This fact may reflect the characteristic of fall injury among geriatric patients, which is significantly different from the traffic-accident-induced injury prevalent in young adults presenting with lower GCS. Another risk factor for mortality discovered by logistic regression was mechanical ventilation. The incidence of receiving mechanical ventilation in non-survivors was 67.6%, which was significantly higher than the 25.9% of survivors. Mechanical ventilation is commonly used to assist breathing for TBI patients with respiratory failure, pulmonary infection, or chest trauma. These patients have worse organ function, higher injury severity and higher risk of a poor outcome.

Finally, abnormal body temperature is prevalent in TBI patients [34]. One previous study found that both elevated temperature and low temperature immediately after prehospital transport were independently associated with higher mortality and with increased length of hospital stay [35]. Elevated temperature after TBI may be caused by a series of factors, such as infection and overactivated sympathetic activity, which may be both associated with poor prognosis.

In addition to factors discovered by the logistic regression, Random Forest and Adaboost algorithms also confirmed several other important factors, including systolic blood pressure, diastolic blood pressure, red cell distribution width, and platelet, based on their contribution degrees to the prediction. The hypotension and even shock status reflected by low blood pressure undoubtedly promote the deterioration of organ function and unfavorable outcomes. Additionally, unstable control of blood pressure and high blood pressure variability would cause the deviation from optimal cerebral perfusion pressure [36]. As a key component of the coagulation system, the platelet has been testified regulating neuroinflammation and restoring blood brain barrier integrity after TBI [37]. Furthermore, platelet dysfunction has been confirmed as one of coagulopathy etiologies after TBI and associated with poor outcomes [38,39]. Finally, previous studies showed red cell distribution width to platelet ratio is a reliable prognostic marker of TBI [40,41].

In our study, the neurosurgical operation did not show an independent association with the mortality of TBI patients analyzed by the logistic regression. Additionally, it did not rank within the top 20 regarding the feature importance of Adaboost and Random Forest. Actually, it is still debated whether conservative or aggressive treatment should be provided for geriatric TBI patients. Although many centers have adopted the conservative treatment for geriatric TBI in the past years, increasing evidence supports the benefit of surgical operation for geriatric TBI. One Japanese study found surgical operation was associated with better functional outcomes and lower mortality of geriatric TBI patients with subdural hematoma and GCS ≥ 6 [8]. The effect of surgical management upon geriatric TBI may depend on many factors, such as injury severity, emergence of symptoms, size and location of hematoma mass, surgical options, physical state and comorbidities of patients. It would be worthwhile to design and perform randomized controlled trials to explore the benefit of surgical management for specific geriatric TBI patients in the future.

It is generally recognized that the prognosis of geriatric TBI is poorer than in young adults with TBI. But there is insufficient literature and studies specially focusing on multiple fields of geriatric TBI patients, including risk evaluation, treatments, prognosis and rehabilitation. Up to now, there has not been a widely acknowledged prognostic risk assessment tool for the geriatric TBI. Previous studies have explored the prognostic value of International Mission for Prognosis and Analysis of Clinical Trials in TBI (IMPACT) score and Corticosteroid Randomization after Significant Head Injury (CRASH) score in geriatric TBI patients [33,42,43,44]. One of them found IMPACT showed moderate discrimination and slight overestimation of the actual outcome for geriatric TBI [42]. And another confirmed that CRASH was an effective prognostic tool for geriatric TBI and it showed no difference of performance between geriatric patients and young patients [44]. However, the small sample size and the highly specialized TBI population of these studies limit the reliability of conclusions. Some studies have utilized logistic regression to develop prognostication tools specific to geriatric TBI, based on multiple factors such as age, GCS, hypotension, Charlson Comorbidity Index and ISS [8,22,28]. Previous studies found machine learning algorithms-based models performed well on the prediction of prognosis in many kinds of neurosurgical patients, such as aneurysmal subarachnoid hemorrhage, and intracerebral hemorrhage [45,46,47]. Additionally, some studies exploring the prognostic value of machine learning in pediatric TBI found machine learning performed better than conventional statistical models and CT scores in predicting outcomes [48,49], while there is still no study exploring the prognostic value of machine learning algorithms in geriatric TBI patients. The results of our study show that machine learning algorithms did not perform worse than the logistic regression, and even show slightly higher accuracy than the logistic regression. The greater statistical difference needs to be verified in a study with a larger sample size. Adaboost and Random Forest showed the best accuracy among several machine learning algorithms adopted in our study. Based on the bagging method, Random Forest is a classifier containing multiple decision trees. Its output category is determined by the mode of individual trees’ category output. There are several advantages of Random Forest, including high accuracy, fast running speed on large datasets, and maintained accuracy in the case of a large part of missing data [50,51]. The Adaboost algorithm is an effective and practical boosting algorithm. Its algorithmic principle is to select weak classifiers with the smallest weight coefficient from the trained weak classifiers by adjusting the sample weight and the weight of the weak classifier, and then combine the two into a final strong classifier [52].

This study has several limitations. Firstly, TBI patients analyzed in this study were identified in the MIMIC-III, which is a freely available intensive care database produced by a hospital in Boston, United States with large sample size. Geriatric TBI patients from this database are mainly classified into mild to moderate brain injury with GCS quartiles of 7 and 15. Therefore, selection bias could not be avoided and future studies mainly including moderate to severe geriatric TBI patients conducted in other medical centers may offer external support to our findings. Secondly, the prognosis is different between mild and moderate to severe TBI patients. Developing machine learning based prognostic models for these two groups of TBI respectively may be more individualized and accurate. Thirdly, though many clinical factors and laboratory indexes have been brought into this study, there are still some risk factors of poor prognosis that have not been collected, such as antiplatelet drugs, anticoagulants and comorbidities excepting for diabetes mellitus and hypertension. Fourthly, several previously developed scores were not recorded and compared with our predictive models such as International Mission for Prognosis and Analysis of Clinical Trials in TBI (IMPACT), Corticosteroid Randomization after Significant Head Injury (CRASH) and Marshall CT score. Finally, the only outcome of this study was 30-day mortality, we did not collect functional outcome and cognitive status which were important measures for evaluating prognosis of geriatric patients due to the nature of the database study.

## 5. Conclusions

Adaboost and Random Forest performed slightly better than the logistic regression on predicting mortality of geriatric TBI patients. Future works could be focused on developing practical application software utilizing these algorithms in portable electronic equipment to quickly evaluate prognosis of geriatric TBI.

## Figures and Tables

**Figure 1 brainsci-13-00094-f001:**
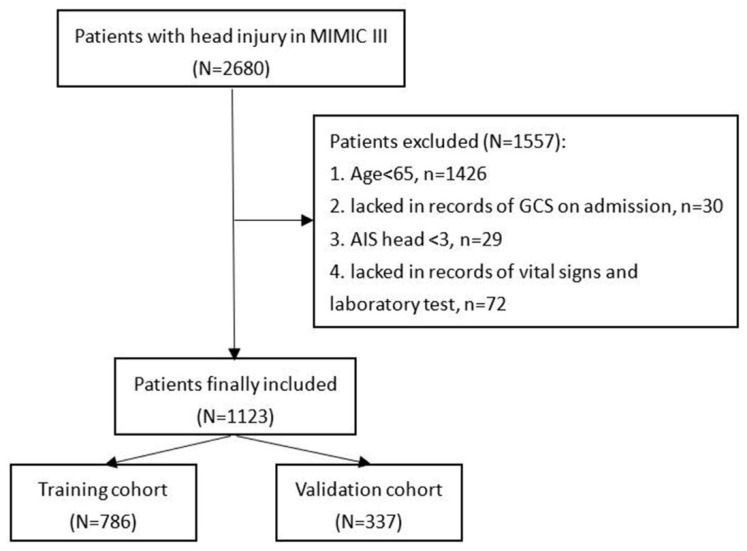
Flowchart of patients’ inclusion.

**Figure 2 brainsci-13-00094-f002:**
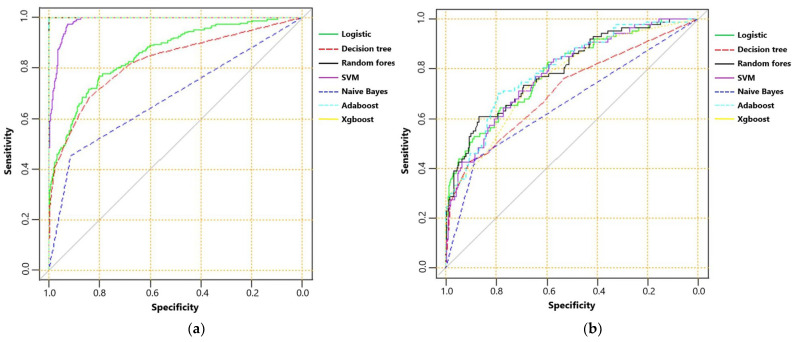
(**a**) The receiver operating characteristic curve of machine learning algorithms for predicting mortality in training set; and (**b**) receiver operating characteristic curve of machine learning algorithms for predicting mortality in testing set.

**Figure 3 brainsci-13-00094-f003:**
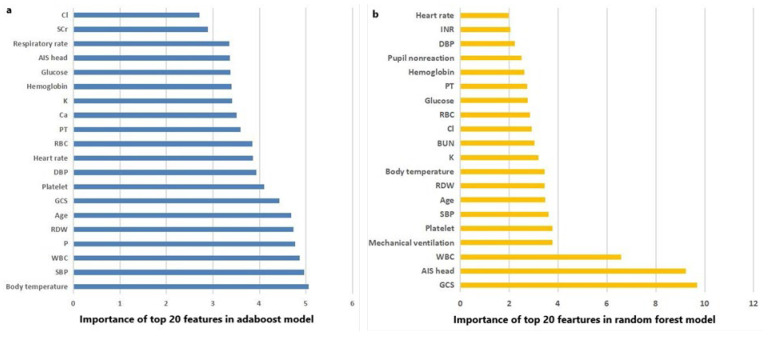
(**a**) The importance of the top 20 features in the Adaboost model; and (**b**) the importance of the top 20 features in the Random Forest model.

**Table 1 brainsci-13-00094-t001:** Baseline characteristics of geriatric TBI patients in MIMIC-III.

Variables	Overall Patients (*n* = 1123)	Survivors (*n* = 845, 75.2%)	Non-Survivors (*n* = 278, 24.8%)	*p*
Age (year)	81.0 (74.6–86.6)	80.7 (74.0–85.9)	82.2 (76.5–87.9)	**0.010**
Male gender (%)	571 (50.8%)	426 (50.4%)	145 (52.2%)	0.663
Diabetes (%)	258 (23.0%)	185 (21.9%)	73 (26.3%)	0.156
Hypertension (%)	630 (56.1%)	494 (58.5%)	136 (48.9%)	**0.007**
Systolic blood pressure (mmHg)	137 (121–152)	138 (123–153)	135 (114–150)	**0.014**
Diastolic blood pressure (mmHg)	65 (53–76)	65 (54–76)	63 (52–75)	0.099
Heart rate (s^−1^)	80 (70–91)	80 (70–91)	81 (70–93)	0.171
Respiratory rate (s^−1^)	18 (15–20)	18 (15–20)	18 (15–21)	0.400
Body temperature (℉)	97.9 (96.9–99.0)	98.0 (97.1–99.0)	97.6 (96.4–98.7)	**<0.001**
SpO_2_ (%)	98 (96–100)	98 (96–100)	99 (97–100)	**<0.001**
Pupillary nonreactivity (size, %)				**<0.001**
None	969 (86.3%)	773 (91.5%)	196 (70.5%)	
One size	64 (5.7%)	42 (5.0%)	22 (7.9%)	
Two size	90 (8.0%)	30 (3.6%)	60 (21.6%)	
GCS	14 (7–15)	14 (10–15)	7 (5–13)	**<0.001**
AIS face	0	0	0	0.315
AIS head	4 (3–4)	4 (3–4)	4 (4–5)	**<0.001**
AIS chest	0	0	0	**0.029**
AIS abdomen	0	0	0	0.870
AIS surface	0	0	0	0.158
AIS limb	0	0	0	0.728
ISS	16 (16–20)	16 (16–17)	16 (16–25)	**<0.001**
Epidural hematoma (%)	165 (14.7%)	106 (12.5%)	59 (21.2%)	**0.001**
Subdural hematoma (%)	696 (62.0%)	533 (63.1%)	163 (58.6%)	0.210
Subarachnoid hemorrhage (%)	403 (35.9%)	289 (34.2%)	114 (41.0%)	**0.048**
Intracerebral hemorrhage (%)	185 (16.5%)	135 (16.0%)	50 (18.0%)	0.490
White blood cell (10^9^/L)	10.80 (8.10–14.10)	10.30 (7.70–13.40)	12.65 (9.53–16.43)	**<0.001**
Platelet (10^9^/L)	216 (173–267)	220 (176–269)	206 (165–260)	**0.030**
Red blood cell (10^9^/L)	3.96 (3.58–4.36)	3.97 (3.59–4.37)	3.90 (3.42–4.34)	0.100
Red cell distribution width (%)	13.9 (13.2–14.9)	13.8 (13.2–14.7)	14.1 (13.4–15.3)	**<0.001**
Hemoglobin (g/dL)	12.2 (10.9–13.4)	12.3 (11.1–13.5)	12.0 (10.5–13.2)	**0.010**
Glucose (mg/dL)	137 (113–173)	132 (110–163)	160 (128–192)	**<0.001**
Blood urea nitrogen (mg/dL)	21 (16–28)	21 (16–27)	23 (17–32)	**<0.001**
Serum creatinine (mg/dL)	1.00 (0.80–1.30)	1.00 (0.80–1.20)	1.10 (0.90–1.40)	**<0.001**
Sodium (mmol/L)	139 (137–141)	139 (137–141)	139 (137–142)	0.384
Potassium (mmol/L)	4.00 (3.70–4.50)	4.00 (3.70–4.40)	4.00 (3.60–4.50)	0.548
Phosphorus (mmol/L)	3.20 (2.70–3.70)	3.20 (2.80–3.70)	3.20 (2.70–3.80)	0.853
Calcium (mmol/L)	8.50 (7.35–9.00)	8.50 (7.70–9.10)	8.30 (1.18–9.00)	**0.002**
Magnesium (mmol/L)	1.90 (1.70–2.10)	1.90 (1.70–2.10)	1.90 (1.60–2.10)	0.519
Chloride (mmol/L)	103 (100–106)	103 (100–106)	103 (100–107)	0.093
Anion gap (mmol/L)	15 (13–17)	15 (13–17)	15 (14–17)	**0.002**
Prothrombin time (s)	13.10 (12.40–15.00)	13.00 (12.30–14.70)	13.40 (12.62–15.85)	**<0.001**
International normalized ratio	1.10 (1.00–1.40)	1.10 (1.00–1.30)	1.20 (1.10–1.50)	**<0.001**
Mechanical ventilation (%)	407 (36.2%)	219 (25.9%)	188 (67.6%)	**<0.001**
Neurosurgical operation (%)	269 (24.0%)	205 (24.3%)	64 (23.0%)	0.735
30-day mortality (%)	278 (24.8%)	0 (0.0%)	278 (100.0%)	**<0.001**
Length of hospital stay (day)	7 (4–12)	7 (4–12)	6 (3–10)	**<0.001**

SpO_2_, pulse oxygen saturation; GCS, Glasgow Coma Scale; AIS, Abbreviated Injury Score; ISS, Injury Severity Score. The bold value indicated *p* < 0.05.

**Table 2 brainsci-13-00094-t002:** Performance of machine learning algorithms for predicting 30-day mortality in the training set and the testing set.

Training Set	AUC	95% CI	Sensitivity	Specificity	Accuracy	F Score
Decision tree	0.825	0.789–0.861	0.686	0.840	0.803	0.628
Random Forest	1.000	1.000	1.000	1.000	1.000	1.000
SVM	0.985	0.979–0.991	0.974	0.928	0.938	0.884
Naïve Bayes	0.684	0.647–0.721	0.455	0.913	0.802	0.527
Logistic	0.859	0.828–0.890	0.77	0.802	0.793	0.643
Adaboost	1.000	1.000	1.000	1.000	1.000	1.000
XGboost	1.000	1.000	1.000	0.998	1.000	1.000
Testing set	AUC	95% CI	Sensitivity	Specificity	Accuracy	F score
Decision Tree	0.712	0.647–0.777	0.425	0.908	0.783	0.503
Random Forest	0.795	0.739–0.851	0.609	0.868	0.801	0.613
SVM	0.785	0.730–0.840	0.713	0.712	0.712	0.561
Naïve Bayes	0.658	0.602–0.715	0.437	0.880	0.766	0.490
Logistic	0.792	0.736–0.848	0.644	0.784	0.745	0.561
Adaboost	0.799	0.746–0.853	0.701	0.792	0.769	0.610
XGboost	0.766	0.709–0.823	0.724	0.680	0.691	0.548

AUC, area under the receiver operating characteristic curve; SVM, support vector machine; Adaboost, adaptive boost; XGboost, extreme gradient boost.

**Table 3 brainsci-13-00094-t003:** Univariate and multivariate logistic regression analysis of risk factors for 30-day mortality in training set.

Variables	Univariate Logistic Regression Analysis			Multivariate Logistic Regression Analysis		
	OR	95% Cl	*p*	OR	95% Cl	*p*
Age	1.031	1.009–1.054	**0.006**	1.054	1.023–1.087	**0.001**
Male gender	1.044	0.753–1.447	0.796			
Diabetes	1.370	0.940–1.996	0.101			
Hypertension	0.774	0.558–1.074	0.125			
Systolic blood pressure	0.992	0.985–0.998	**0.013**	1.002	0.994–1.010	0.667
Diastolic blood pressure	0.991	0.982–1.001	0.077			
Heart rate	1.007	0.997–1.016	0.171			
Respiratory rate	0.991	0.960–1.024	0.595			
Body temperature	0.783	0.709–0.866	**<0.001**	0.825	0.728–0.934	**0.002**
SpO_2_	1.052	0.993–1.115	0.084			
Pupillary nonreactivity			**<0.001**			**0.001**
None	1.000	[Reference]		1.000	[Reference]	
One size	1.930	0.995–3.743	0.052	1.509	0.668–3.410	0.322
Two size	9.797	5.546–17.305	**<0.001**	3.745	1.818–7.716	**<0.001**
GCS	0.786	0.754–0.820	**<0.001**	0.888	0.831–0.948	**<0.001**
AIS face	0.899	0.700–1.154	0.403			
AIS head	2.767	1.993–3.841	**<0.001**	2.383	1.309–4.339	**0.004**
AIS chest	1.202	1.044–1.384	**0.010**	1.071	0.776–1.477	0.678
AIS abdomen	1.041	0.767–1.414	0.796			
AIS surface	0.669	0.291–1.542	0.346			
AIS limb	1.057	0.869–1.287	0.579			
ISS	1.069	1.045–1.094	**<0.001**	0.980	0.919–1.044	0.530
Epidural hematoma	1.776	1.154–2.734	**0.009**	1.419	0.788–2.556	0.244
Subdural hematoma	0.791	0.567–1.103	0.166			
Subarachnoid hemorrhage	1.204	0.859–1.687	0.282			
Intracerebral hemorrhage	1.317	0.871–1.992	0.192			
White blood cell	1.097	1.063–1.133	**<0.001**	1.077	1.039–1.117	**<0.001**
Platelet	1.000	0.998–1.002	0.897			
Red blood cell	0.843	0.658–1.080	0.176			
Red cell distribution width	1.121	1.022–1.230	**0.015**	1.118	0.981–1.273	0.093
Hemoglobin	0.903	0.832–0.981	**0.015**	0.916	0.816–1.029	0.139
Glucose	1.006	1.003–1.008	**<0.001**	1.002	0.999–1.005	0.239
Blood urea nitrogen	1.013	1.002–1.024	**0.026**	1.002	0.986–1.018	0.831
Serum creatinine	1.183	0.979–1.429	0.081			
Sodium	1.020	0.983–1.059	0.302			
Potassium	0.955	0.751–1.215	0.710			
Phosphorus	1.011	0.833–1.228	0.909			
Calcium	0.946	0.902–0.993	**0.025**	1.113	1.038–1.193	**0.003**
Magnesium	0.916	0.541–1.553	0.745			
Chloride	1.048	1.016–1.082	**0.003**	1.025	0.983–1.067	0.245
Anion gap	1.054	1.002–1.109	**0.041**	1.048	0.975–1.125	0.201
Prothrombin time	1.018	0.997–1.038	0.089			
International normalized ratio	1.185	1.000–1.404	0.050			
Mechanical ventilation	5.768	4.053–8.208	**<0.001**	3.542	2.012–6.238	**<0.001**
Neurosurgical operation	0.992	0.676–1.457	0.969			

OR, odds ratio; CI, confidence interval; SpO2, pulse oxygen saturation; GCS, Glasgow Coma Scale; AIS, Abbreviated Injury Score; ISS, Injury Severity Score. The bold value indicated *p* < 0.05.

## Data Availability

The datasets used for the current study are available from the corresponding author on reasonable request.

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
