# Peer review of "Prediction of Mortality in Geriatric Traumatic Brain Injury Patients Using Machine Learning Algorithms"

_brainsci, 2023, doi:10.3390/brainsci13010094_

Round 1
Reviewer 1 Report
The Authors studied the possibility of predicting the outcome (survival/death) of geriatric patients with TBI. The subject is interesting and the statistical analysis is accurate. Nevertheless there are several limitations, in my opinion.
- All the patients are grouped together. I think that it would be better to separate mild TBI from severe TBI.
- It appears from the analysis that neurosurgical treatment is not related to outcome, but this statistical result should be discussed further. It depends in my opinion from the type of intervention and obviuosly from the cause that led to the surgical treatment (or to avoid the surgical treatment, as possible in geriatric patients)
- Body temperature is a significant risk for death, this should be discusses. -- - Mechanical ventilation: this should be considered in any phase of the hospitalization or only at the beginning.
The major flaw of the work is, in my opinion, in the design. When considering all variables in the same algorithm, the significance of each variable is difficult to interpret. This makes clinical use of such algorithm not feasible. The Authors should clarify the objective of the study. I think this could be the basement for subsequent studies that should be more targeted, but, in my opinion, not for clinical use.
Author Response
1. All the patients are grouped together. I think that it would be better to separate mild TBI from severe TBI.
Response: Thanks for this valuable suggestion. The prognosis of mild TBI and non-mild TBI is indeed different. Developing respective prognostic models for these two groups of patients may be more personalized and accurate. While after check, we found there were 477 moderate to severe TBI patients and 646 mild TBI patients in our study. If we divided them into two groups, the sample sizes of these two groups may not be enough to develop stable and accurate machine learning based prognostic models. Because the accuracy and stability of machine learning classification may depend on the relatively huge sample size of participants. Therefore, to fully highlight the advantages of machine learning classification based on a huge sample size, we did not divide patients into mild and non-mild TBI. Certainly, future studies with larger sample sizes could be performed to develop respective machine learning predictive models targeted at mild and moderate to severe TBI. Additionally, the lack of separation of mild from severe TBI had been stated in the limitation part of our manuscript.
2. It appears from the analysis that neurosurgical treatment is not related to outcome, but this statistical result should be discussed further. It depends in my opinion from the type of intervention and obviuosly from the cause that led to the surgical treatment (or to avoid the surgical treatment, as possible in geriatric patients)
Response: Thanks for this suggestion. We have discussed the neurosurgical treatment in our manuscript as following:” Actually, it is still debated whether conservative or aggressive treatment should be provided for geriatric TBI patients. Although many centers adopt the conservative treatment for geriatric TBI in the past years, increasing evidence supports the benefit of surgical operation for geriatric TBI. One Japanese study found surgical operation was associated with better functional outcome and lower mortality of geriatric TBI patients with subdural hematoma and GCS ≥6 [8]. The effect of surgical management for geriatric TBI may depend on many factors such as the injury severity, emergence of symptoms, size and location of hematoma mass, surgical options, physical state and comorbidities of patients. It is worthwhile to design and perform randomized controlled trials to explore the benefit of surgical management for specific geriatric TBI patients in the future.”
3. Body temperature is a significant risk for death, this should be discusses. -- - Mechanical ventilation: this should be considered in any phase of the hospitalization or only at the beginning.
Response: Thanks for this suggestion. The discussion about body temperature has been added in our revised manuscript as following” Finally, the abnormal body temperature is prevalent in TBI patients [34]. One previous study found both elevated temperature and low temperature immediately after prehospital transport were independently associated with higher mortality and with increased length of hospital stay [35]. Elevated temperature after TBI may be caused by a series of factors such as infection, overactivated sympathetic activity, which may be both associated with poor prognosis.”
The mechanical ventilation included in our study was recorded during the hospitalization. As we stated:” The mechanical ventilation is commonly used to assist breathing for TBI patients with respiratory failure, pulmonary infection or chest trauma. These patients have worse organ function, higher injury severity and higher risk of poor outcome.“ The mechanical ventilation is usually used at the beginning after TBI, while it would also be started a period after admission due to the deteriorating respiratory condition.
4. The major flaw of the work is, in my opinion, in the design. When considering all variables in the same algorithm, the significance of each variable is difficult to interpret. This makes clinical use of such algorithm not feasible. The Authors should clarify the objective of the study. I think this could be the basement for subsequent studies that should be more targeted, but, in my opinion, not for clinical use.
Response: Thanks for this suggestion. Though Adaboost, random forest were confirmed perform well in predicting mortality of geriatric TBI patients in our study. We did not develop conveniently clinical use prognostication tool for physicians. We just verify the efficiency of Adaboost, random forest on evaluating the risk of mortality in geriatric TBI. Compared with the logistic regression, machine learning algorithms based models are not visible and readily used as a visual score. The possibility of outcome evaluated by machine learning algorithms must be calculated using programs in computers. Therefore, as we stated in the conclusion part of our manuscript” Future works could be focused on developing practical application software utilizing these algorithms in portable electronic equipment to quickly evaluate prognosis of geriatric TBI.”
Reviewer 2 Report
The authors aimed to compare the prognostic value of different machine learning models for geriatric (≥65 years) TBI patients. These patients were derived from the Medical Information Mart for Intensive Care-III (MIMIC-III) database. Among different machine learning algorithms, adaboost (AUC: 0.799) and random forest (AUC: 0.795) performed slightly better than the logistic regression (AUC: 0.792) on predicting mortality of geriatric TBI patients in the testing set. The authords concluded that adaboost, random forest and logistic regression all performed well in predicting mortality, and that prognostication tools utilizing these algorithms are helpful to evaluate the risk of poor outcome and adopt personalized therapeutic options. As there are no studies using machine learning algorithms to evaluate the prognosis of geriatric TBI, this is the first one.
The manuscript is well-structured. 17/40 references were within the last 5 years. The experimental design was appropriate for the aim. Results are mostly reproducible based on the details given in the methods section, minor edits needed. Figures and tables are appropriate, minor edits needed. The data interpreted appropriately, minor edits needed. Conclusions need some edits.
Below are some comments/edits needed:
Line34: I would not refer to that as a “severe problem confronted”, maybe a “challenge”.
Line40+43: Are these blank parts of sentences a mistake?
Line 81: Please give the full name of the abbreviations GCS and AIS when they first appear in the text, and maybe give a reference and a sentence of what AIS tests.
Line 91: A reference for ISS would be great.
Line 94: If word limit allows and it is easy, an inclusion of ICD9 codes for each variable mentioned in lines 92+93 would enhance the reproducibility of this study.
Line 99, 208, tables: Does “neurosurgery” mean any neurosurgical intervention? Or admission? Or surgical operation?
Lines 113-115 I could not find any variable given in mean (SD). Were all variables abnormally distributed?
Line 111: Is there any rationale behing the 70/30? Why not 80/20? Is there literature on that, or they usually choose the number arbitarily?
Line 111, 151, 152: Was the logistic regression, uni and multivariable, used in all cohort, or separately in the training and testing cohort? Please mention that in methods and results.
Line 118: Could you briefly mention what F1 score represents?
Line 127: Is “complicated” there by mistake?
Line 127-136: You can mention SpO2 too.
Line 150-151: Do you mean “the most important”, most predictive? Is it for both training and testing set? Because there are only 2 and not 4 figures. Maybe you should include some info on how importance is defined. Is it OR? Also mention maybe the top 3 most important predictors in the text too.
Line 150-151: Are these predictors independent? Does this work like a multivariable logistic regression?
Figures 3A,B: What does the numbers 0 to 6 in X line mean? Does low or high mean more predictive? Also abbreviations should be included in this figure and maybe the 20 variables should be more visible (larger font).
Table 3: Some variables were significantly different in table 1 (like hypertension, SpO2 etc) but were not significant in the logistic regression. Did you or should you include them in the multivariate model?
Line 192: Should “25.9% of non-survivors.” be “25.9% of survivors.”?
Line 220: It would be great to include a couple of references here.
Line 234-247: This could be a separate paragraph. How have these machine learning models helped TBI in young patients or other neurosurgical diseases/ other specialties eg general surgery? Are these studies similar to your paper but for another disease? If yes please include some examples. Why would someone use machine learning methods if we have logistic regression? How do these new predictors differ from the predictors found with a multivariate logistic regression, and why did not logistic regression find them? Do the common predictors found between random/adaboost/regression are “stronger”? Please give some examples.
Line 266-267 “Practical programs utilizing these algorithms in portable electronic equipment” What does “practical programs” mean? Aren’t these algorithms just to find predictors in studies? Do they work somehow in real time in “portable electronic equipment”? Or does that refer to scales derived from such studies? This sentence is strange please clarify what you mean.
Line 279-281: Did this study need IRB approval? Please mention it, even if approval was not needed because the database is free with deidentified data.
Author Response
Line34: I would not refer to that as a “severe problem confronted”, maybe a “challenge”.
Response: Thanks for this suggestion. We have revised as you recommended.
Line40+43: Are these blank parts of sentences a mistake?
Response: Thanks for this suggestion. We have delated the blank parts.
Line 81: Please give the full name of the abbreviations GCS and AIS when they first appear in the text, and maybe give a reference and a sentence of what AIS tests.
Response: Thanks for this suggestion. We have revised as you recommended. The reference of AIS has been added.
Line 91: A reference for ISS would be great.
Response: Thanks for this suggestion. The reference for ISS has been added.
Line 94: If word limit allows and it is easy, an inclusion of ICD9 codes for each variable mentioned in lines 92+93 would enhance the reproducibility of this study.
Response: Thanks for this suggestion. The ICD-9 codes for epidural hematoma, subdural hematoma and subarachnoid hemorrhage are relatively easy to display while the ICD-9 codes for intracerebral hemorrhage are too many to display. If you consider those ICD-9 codes are necessary, we would supply them in supplementary materials.
Line 99, 208, tables: Does “neurosurgery” mean any neurosurgical intervention? Or admission? Or surgical operation?
Response: Thanks for this suggestion. We have changed the neurosurgery to neurosurgical operation in our manuscript which means any neurosurgical operations during hospitalizations.
Lines 113-115 I could not find any variable given in mean (SD). Were all variables abnormally distributed?
Response: Thanks for this comment. Actually, variables included in this study were all non-normally distributed. Therefore, all variables were shown as median (interquartile) but not the mean (SD).
Line 111: Is there any rationale behing the 70/30? Why not 80/20? Is there literature on that, or they usually choose the number arbitarily?
Response: Thanks for this comment. The ratio of 7:3 and 8:2 are both conventionally used to divide overall dataset into the training set and testing set during the process of developing machine learning models. Screening previous studies utilizing machine learning, we found 7:3 may be more prevalently used. Additionally, compared with 8:2, the ratio of 7:3 may reduce the possibility of imbalance between the training set and testing set.
Line 111, 151, 152: Was the logistic regression, uni and multivariable, used in all cohort, or separately in the training and testing cohort? Please mention that in methods and results.
Response: Thanks for this suggestion. We performed the univariate and multivariate logistic regression in the training set. We have mentioned this point in the revised manuscript.
Line 118: Could you briefly mention what F1 score represents?
Response: Thanks for this suggestion. F1 score is an index reflecting the accuracy of classification task and usually used in evaluating the accuracy of machine learning algorithms. It is calculated as the harmonic average of the precision rate and recall rate. The precision rate is calculated as TP/(TP+FP) and the recall rate is calculated as TP/(TP+FN). TP-True positive; FP-false positive; FN-false negative. A brief description about the F1 score has been added in the revised manuscript.
Line 127: Is “complicated” there by mistake?
Response: Thanks for this suggestion. We have delated the “complicated”.
Line 127-136: You can mention SpO2 too.
Response: Thanks for this suggestion. We have added the description of SpO2 as you suggested.
Line 150-151: Do you mean “the most important”, most predictive? Is it for both training and testing set? Because there are only 2 and not 4 figures. Maybe you should include some info on how importance is defined. Is it OR? Also mention maybe the top 3 most important predictors in the text too.
Response: Thanks for this comment. Figure 3a, 3b showed the importance of features in the Adaboost and random forest in the training set. The high value in X line means more important in influencing the prognosis of geriatric TBI. The top 3 most important features in the Adaboost and random forest has been added in the text of revised manuscript. The importance of features was not evaluated by OR. The OR value is a reflection of association strength between factors and outcomes in the logistic regression but not in the machine learning algorithms including the Adaboost and random forest. For example, the importance of features in the random forest is evaluated by the percentage of increase of mean square error (Increase in MSE(%)).
Line 150-151: Are these predictors independent? Does this work like a multivariable logistic regression?
Response: Thanks for this comment. The machine learning algorithms including the Adaboost and random forest works differently from the logistic regression. The advantage of machine learning is to identify factors significantly influencing outcomes in complexed high dimensional dataset but not to discover independent factors. These predictors may not be independent.
Figures 3A,B: What does the numbers 0 to 6 in X line mean? Does low or high mean more predictive? Also abbreviations should be included in this figure and maybe the 20 variables should be more visible (larger font).
Response: Thanks for this comment. The higher number in X line means more important in influencing the prognosis. The figure 3a, 3b has been updated to be more visible.
Table 3: Some variables were significantly different in table 1 (like hypertension, SpO2 etc) but were not significant in the logistic regression. Did you or should you include them in the multivariate model?
Response: Thanks for this suggestion. The table 1 showed the information of all included patients while the logistic regression showed the association in the training cohort. The univariate logistic regression showed hypertension and SpO2 were not associated with the outcome in the training cohort. Therefore, we did not include them in the multivariate model.
Line 192: Should “25.9% of non-survivors.” be “25.9% of survivors.”?
Response: Thanks for this remind. We have updated the result as you mentioned.
Line 220: It would be great to include a couple of references here.
Response: Thanks for this suggestion. We have added references for those two algorithms.
Line 234-247: This could be a separate paragraph. How have these machine learning models helped TBI in young patients or other neurosurgical diseases/ other specialties eg general surgery? Are these studies similar to your paper but for another disease? If yes please include some examples. Why would someone use machine learning methods if we have logistic regression? How do these new predictors differ from the predictors found with a multivariate logistic regression, and why did not logistic regression find them? Do the common predictors found between random/adaboost/regression are “stronger”? Please give some examples.
Response: References of machine learning models in other kinds of neurosurgical patients has been added in our revised manuscript as following:”Previous studies found machine learning algorithms-based models performed well on the prediction of prognosis in many kinds of neurosurgical patients such as aneurysmal subarachnoid hemorrhage, intracerebral hemorrhage [43-45]. Additionally, some studies exploring the prognostic value of machine learning in pediatric TBI found machine learning performed better than conventional statistical models and CT scores in predicting outcomes [46, 47].” The machine learning algorithms-based models may perform better than the logistic regression in some settings. It performs better in analyzing nonlinear correlations and handling massive high-dimensional datasets which is similar to the real world clinical data. New predictors may not be totally consistent with those discovered by the logistic regression. Because the logistic regression discovered independent risk factors but the machine learning is not designed to discover independent risk factors. And factors in the machine learning are not necessarily independent. The association between factors and outcomes in the machine learning may be high-dimensional. There is no evidence to support the idea that common predictors found between the machine learning and logistic regression are “stronger”.
Line 266-267 “Practical programs utilizing these algorithms in portable electronic equipment” What does “practical programs” mean? Aren’t these algorithms just to find predictors in studies? Do they work somehow in real time in “portable electronic equipment”? Or does that refer to scales derived from such studies? This sentence is strange please clarify what you mean.
Response: Thanks for this suggestion. We have updated the sentence as following ”Future works could be focused on developing practical application software utilizing these algorithms in portable electronic equipment to quickly evaluate prognosis of geriatric TBI.” Our study found the Adaboost and random forest performed well on predicting mortality of geriatric TBI. While we just verify the prognostic value of these two algorithms using R software but not developing a practical application software which could be conveniently used by clinicians to evaluate prognosis of geriatric TBI. It is worthwhile to develop a practical application software utilizing these algorithms to evaluate prognosis in future work.
Line 279-281: Did this study need IRB approval? Please mention it, even if approval was not needed because the database is free with deidentified data.
Response: Thanks for this remind. This study did not need IRB approval. Because this study was conducted collecting data from a free database with deidentified data. This fact has been stated in our revised manuscript.